# Plant genotype determines biomass response to flooding frequency in tidal wetlands

Svenja Reents[1], Peter Mueller[2], Hao Tang[1], Kai Jensen[1], Stefanie Nolte[3,4]

[1]Applied Plant Ecology, Institute of Plant Science and Microbiology, Universität Hamburg, Hamburg, 22609, Germany
[2]Smithsonian Environmental Research Center, Edgewater, Maryland, 21087, United States
[3]School of Environmental Sciences, University of East Anglia, Norwich, NR4 7TJ, United Kingdom
[4]Centre for Environment, Fisheries and Aquaculture Science, Lowestoft, NR33 0HT, United Kingdom

*Correspondence to*: Svenja Reents (svenja.reents@uni-hamburg.de)

**Abstract.** The persistence of tidal wetland ecosystems like salt marshes is threatened by human interventions and climate change. Particularly the threat of accelerated sea level rise (SLR) has recently gained increasing attention by the scientific community. However, studies investigating the effect of SLR on plants and vertical marsh accretion are usually restricted to the species or community level and do not consider phenotypic plasticity or genetic diversity. To investigate the response of genotypes within the same salt-marsh species to SLR, we used two known genotypes of *Elymus athericus* (Link) Kerguélen
(low-marsh and high-marsh genotypes). In a factorial marsh organ experiment we exposed both genotypes to different flooding frequencies and quantified plant growth parameters. With increasing flooding frequency, the low-marsh genotype showed higher aboveground biomass production compared to the high-marsh genotype. Additionally, the low-marsh genotype generally formed longer rhizomes, shoots and leaves, regardless of flooding frequency. Belowground biomass of both genotypes decreased with increasing flooding frequency. We conclude that the low-marsh genotype is better adapted to higher
flooding frequencies through its ability to allocate resources from below- to aboveground biomass. Given the strong control of plant biomass production on salt-marsh accretion, we argue that these findings yield important implications for our understanding of ecosystem resilience to SLR as well as plant-species distribution in salt marshes.

## 1 Introduction

Salt marshes are wetland ecosystems predominantly found along coastlines where they form a transition zone between the
marine and the terrestrial environment. Salt marshes provide important ecosystem services like protection of coastlines against storm surges by wave attenuation (Möller et al., 2014), supply of nursery grounds for commercially important fish (Bolle et al., 2009) and mitigation of climate change by long-term carbon sequestration (McLeod et al., 2011). However, human interventions such as land reclamation, eutrophication and climate change threaten the persistence of salt marshes, causing loss rates of 1-2 % of the global area per year (Duarte et al., 2008). Particularly the threat of accelerated sea level rise (SLR)
has recently gained increasing attention by the scientific community (Fitzgerald et al., 2008; Kirwan and Megonigal, 2013; Schuerch et al., 2018).

Generally, up to a certain locally varying threshold of SLR, salt marshes are able to keep up with rates of SLR through their ability to accrete vertically (Kirwan and Megonigal, 2013). During this process, salt-marsh plants act as 'ecosystem engineers' because their aboveground biomass reduces water flow velocity and hydrodynamic forces, which results in a decrease in the

sediment-loading capacity of the water and an increase in sediment settlement (Morris et al., 2002; Yang, 1998). Furthermore, a high belowground biomass production and decreased decomposition rates in oxygen-deficient soils lead to an accumulation of organic matter and thereby expansion of soil volume. Whether above- or belowground processes predominantly contribute to vertical accretion, depends on the marsh type (i.e. minerogenic vs. organogenic) and often depends on local tidal amplitude (Allen, 2000; Kirwan and Megonigal, 2013; Nolte et al., 2013b). Yet, in both cases, biomass production of the marsh vegetation

is an important driver of accretion. When accretion rates are too low and threshold values of SLR are exceeded, plant growth is negatively affected, leading to a negative feedback loop, which ensues marsh submergence and finally marsh loss (Chmura, 2013; Kirwan and Guntenspergen, 2012).

Recently, more studies have therefore been focusing on salt-marsh resilience to projected rates of SLR by studying the vegetation response. To examine how vegetation responds to changes in flooding regimes, so called 'marsh organ' experiments

have been proven as convenient and insightful approaches. A marsh organ consists of several mesocosms arranged next to each other and along an elevational gradient. Kirwan and Guntenspergen (2012), for example, placed a marsh organ in a brackish marsh to examine the effect of varying sea levels on plant growth of two marsh species and what possible implications this may have on accretion processes. They observed that marsh elevation within the tidal frame determines whether root biomass increases or decreases with SLR. In contrast to belowground productivity, aboveground biomass response was species

dependent. Numerous other studies confirmed that responses of marsh plants to sea level changes are highly species-specific (Eleuterius and Eleuterius, 1979; Kirwan and Guntenspergen, 2015; Langley et al., 2013; Morris et al., 2013). Therefore, the effect of SLR on plant biomass production, and thus the ability of the ecosystem to accrete vertically, can only be understood if SLR-induced shifts in plant species composition also are taken into account.

However, such studies on the effect of SLR on plants are usually restricted to the species or community level (Kirwan and

Guntenspergen, 2015; Langley et al., 2013; Morris et al., 2013). They usually do not consider the high degree of phenotypic plasticity and genetic diversity within many species, as for instance evident between different locally adapted populations (Valladares et al., 2014). Yet, understanding a species' adaptive genetic plasticity can be crucial to evaluate its response to environmental change (Razgour et al., 2019). Indeed, in some cases environmental change can in fact induce larger variability in plant biomass production within than between species (Beierkuhnlein et al., 2011). We therefore argue, that intraspecific

differences in the biomass response to SLR of salt-marsh plants are likely relevant and require more attention by the scientific community to understand ecosystem resilience.

To investigate the response of genotypes within the same salt-marsh species to SLR, we used two known genotypes of the grass *Elymus athericus* (Link) Kerguélen. This species is widely distributed in NW European salt marshes and usually grows in high-elevated and therefore less-frequently flooded zones of the salt marsh (= high marsh) (Nolte et al., 2019). In the high

marsh, the tall grass forms dense, monospecific stands with a low local plant species diversity (Bakker et al., 2003). Recently,

however, populations of *E. athericus* have been observed spreading into lower and thus more frequently flooded zones of the marsh (= low marsh) (Olff et al., 1997; Veeneklaas et al., 2013). Low- and high-marsh genotypes are visually distinguishable, as the low-marsh genotype develops a specific phenotype different from the high-marsh genotype (Bockelmann et al., 2003). If genotypes respond differently to increased flooding frequencies, for instance in biomass productivity, it might affect salt-marsh responses to SLR.

The aim of this study was to investigate potential adaptations of the low-marsh genotype to increased flooding frequencies, to improve current evaluations of salt-marsh resilience to SLR. To test the hypothesis that the low-marsh genotype performs better at increased flooding frequencies than the high-marsh genotype, which would be reflected in above- and belowground growth parameters like biomass production and shoot, leaf and rhizome lengths, we conducted a factorial marsh organ experiment. We exposed both genotypes of *E. athericus* to three different flooding frequencies and quantified plant growth parameters to compare their performance and assess morphological adaptations of the low-marsh genotype.

## 2 Material and methods

### 2.1 Plant collection and culture

Plants were collected in April 2015 from a salt marsh on the Dutch Island Schiermonnikoog (53°30′N, 6°16′E) from stands that have previously been identified to be dominated by genetically distinct populations of *E. athericus*, i.e. high-marsh (HM) and low-marsh (LM) genotypes (Bockelmann et al., 2003). On Schiermonnikoog, *Elymus athericus* can be found in higher and lower elevated sites, which are inundated 20 – 125 and 90 – 270 times per year, respectively (Bockelmann et al., 2003). Soil salinities range from 22 to 29 ppt (Bolhuis et al., 2013). Plants and soil were extracted in the form of intact sods to keep them alive during transport. In Hamburg, soil was removed and roots were rinsed before both genotypes were planted separately in trays with standardised potting soil. Until the start of the experiment (i.e. for 24 months), plants were kept under identical environmental conditions in a common garden at the Institute of Plant Science and Microbiology. Ramets of these plants were used for this study. In July 2017, single plants of similar biomass were selected based on visual assessment (no obvious outliers), transplanted to separate pots and randomly assigned to the flooding treatments (described in 2.2). Initial shoot length and shoot number was tested for differences between genotypes and flooding frequencies to ensure that results were not biased by unequal plant size at the beginning of the experiment. There were no significant differences regarding shoot length (genotype: $F = 0.787$, $p = 0.380$; flooding frequency: $F = 0.127$, $p = 0.881$; genotype*flooding frequency: $F = 0.231$, $p = 0.795$) and number of shoots (genotype: Wald $= 2.203$, $p = 0.137$; flooding frequency: Wald $= 0.357$, $p = 0.837$; genotype*flooding frequency: Wald $= 0.005$, $p = 0.997$). The pots were 15 cm in diameter, 17 cm in height and had holes in the bottom to facilitate drainage. They were filled with salt marsh soil taken from the salt marsh at Sönke-Nissen-Koog, Germany (54°36'N, 8°49'E) which was sieved (with a 1 cm mesh) and homogenised beforehand (see Nolte et al. (2013a) for soil properties). Eight replicates (i.e. single plants in separate pots) per genotype were assigned to one of three flooding treatments, so that a total number of 48 plants were used in this study.

## 2.2 Experimental set-up

Plants were placed onto three steps (step height: 20 cm) within a tidal-tank facility (Hanke et al., 2015), to create three different flooding frequencies. The tidal tank is located outdoors at the Institute of Plant Science and Microbiology and has a total volume of 6.75 m³ (dimensions: 3 x 1.5 x 1.5 m). A pump was used to fill and empty the tidal tank at regular intervals to mimic tides by alternating between three different maximum water levels. Pots were fully drained between flooding events. Flooding with the respective maximum water level reached 3 cm above soil surface, lasted two hours and took place twice a day. The lowest step was flooded every day, which represented the highest frequency. The flooding of the middle step (moderate flooding frequency) happened weekly, while plants on the highest step were flooded only every two weeks (lowest flooding frequency). Highest and lowest flooding frequency reflect the natural flooding gradient between pioneer zone and high marsh in many NW European salt marshes, including the site where our plants were collected (Bockelmann et al., 2002). A CTD diver combined with a baro diver (Van Essen Instruments, Delft, The Netherlands) was used to monitor flooding cycles. Artificial sea salt (AB Aqua Medic GmbH, Germany) was suspended in tap water to create a salinity of about 20-22 ppt. To minimise the impact of other effects than flooding frequency and genotype, the pots were circulated on each step, other seedlings and algae were removed once a week. Concurrently, water level and salinity were checked as well. The experiment ran for approx. 12 weeks from mid-July to early October 2017.

## 2.3 Measurements

### 2.3.1 Biomass

At the end of the experiment (2nd of October 2017), the plants were harvested, separated into above- and belowground biomass, dried for two days at 70°C and weighed. Belowground biomass was divided into rhizomes and roots and weighed. The length of rhizomes was also recorded. Additionally, above- and belowground biomass were used to calculate the belowground:aboveground ratio.

### 2.3.2 Plant growth

At the beginning and the end of the experiment, plant shoot and leaf length as well as number of shoots and leaves were measured. The difference between both measurements was calculated and designated as Δ. Only living plant material was taken into account and length measurements (leaf and shoot length) were carried out on the longest leaves or shoots.

### 2.4 Statistical analysis

Data were tested for normality by applying the Shapiro-Wilk-Test. Except count data (e.g. number of leaves), all parameters were normally distributed and therefore further analysed applying factorial ANOVAs. Due to the well-balanced study design, potential moderate deviations from homogeneity of variance between groups were considered unimportant for ANOVA testing (Box, 1954; McGuinness, 2002). Each analysis included genotype and flooding frequency as well as their interaction as

explanatory variables. To detect significant differences between treatments (flooding frequency and genotype), post-hoc tests (Tukey's HSD) were conducted. To analyse count data, i.e. number of shoots and number of leaves, generalized linear models (GLM) were applied assuming a Poisson distribution and including the explanatory variables genotype and flooding frequency, as well as two-way interaction effects. Each GLM was checked for overdispersion (Pearson Chi2 dispersion parameter) and was refitted afterwards if necessary, using the standard procedure of the applied program. All statistical analyses were performed using STATISTICA 13 (StatSoft Inc., Tulsa, OK, USA).

## 3 Results

### 3.1 Biomass

Total biomass, defined as the sum of dry above- and belowground plant biomass, differed significantly between genotypes and flooding frequencies (Table 1). In addition, the interaction of both factors showed a significant effect on total biomass. Total biomass production of the high-marsh genotype decreased steadily with increasing flooding frequency, whereas the total biomass of the low-marsh genotype decreased less distinctly (Fig. 1a). The difference between low-marsh and high-marsh genotypes was most pronounced at the highest flooding frequency. In fact, the low-marsh genotype produced almost twice as much total biomass at highest flooding frequency as the high-marsh genotype (LM: 4.61 ± 0.70 g and HM: 2.66 ± 0.52 g, mean ± standard deviation).

Genotype and flooding frequency as well as their interaction had a significant effect on the aboveground biomass production. Aboveground biomass of the high-marsh genotype decreased with increasing flooding frequency from 3.31 ± 0.57 g to 2.03 ± 0.38 g (Fig. 1b). However, aboveground biomass production of the low-marsh genotype remained constant at about 3.46 ± 0.45 g on all flooding frequencies.

In contrast to the genotype-specific aboveground biomass response to flooding, belowground biomass of both genotypes decreased with increasing flooding frequency (Fig. 1c). Results indicate a more pronounced effect of flooding frequency on belowground biomass production compared to the factor genotype (Table 1). Under all flooding frequencies, the low-marsh genotype produced slightly more belowground biomass than the high-marsh genotype (LM: 1.63 ± 0.78 g, HM: 1.30 ± 0.76 g).

Root biomass production (belowground biomass without rhizomes) was significantly affected by flooding frequency (F = 10.69, $p < 0.001$), but did not differ between genotypes. Root biomass decreased with increasing flooding frequency for both genotypes (Fig. 1d). In contrast, biomass of rhizomes was significantly affected by both genotype and flooding frequency (Table 1). Mean rhizome biomass of the low-marsh genotype was higher than of the high-marsh genotype (LM: 0.55 ± 0.50 g, HM: 0.26 ± 0.28 g) with the most pronounced differences on lowest and highest flooding frequency (Fig. 1e). In some cases, the low-marsh genotype formed very long rhizomes (up to 166 cm length, coiled around the soil). On average, rhizomes of the low-marsh genotype were nearly twice as long as those of the high-marsh genotype (LM: 51.43 ± 41.11 cm, HM: 26.63 ± 27.23 cm, Table 2). Genotypes significantly differed in the length of rhizomes (F = 6.102, $p < 0.05$, Table 1).

Belowground:aboveground-ratio was significantly affected by flooding frequency (Table 1) and decreased with increasing flooding frequency (Fig. 1f).

### 3.2 Δ Leaf and shoot length

The increase in shoot and leaf length significantly differed between genotypes (Table 1). Regarding leaf length, the high-marsh genotype showed approximately the same increase on all flooding frequencies (1.9 ± 4.45 cm). The low-marsh genotype had
similar increases of leaf lengths at the lowest flooding frequency but showed pronounced increases of leaf length with increasing flooding frequency (7.03 ± 2.17 cm) (Fig. 2a). Increase in shoot length of the low-marsh genotype was twice as high as that of the high-marsh genotype (LM: 10.78 ± 6.18 cm, HM: 5.57 ± 6.58 cm, Fig. 2b).

### 3.3 Δ Number of leaves and shoots

Neither genotype nor flooding frequency had a significant effect on the increase in number of shoots (Table 1). However, for
the increase in number of leaves, a significant effect of flooding frequency was detected (Wald = 19.69, p < 0.001). With highest flooding frequency, both genotypes produced the lowest number of new leaves (LM: 10.4 ± 4.0, HM: 8.9 ± 2.1; Table 2).

### 4 Discussion

Assessments of plant responses to changed hydrological conditions (e.g. SLR) have thus far focused mainly on comparisons
on species level. However, variability in plant responses within species can be considerably higher than between species (Beierkuhnlein et al., 2011). In this study, we therefore investigated differences in plant response between genotypes of the same species (*Elymus athericus*) to better understand the importance of intraspecific variability for evaluations of future ecosystem functionality and resilience. We found, in line with our hypothesis on biomass production, that the low-marsh genotype performs better than the high-marsh genotype under increased flooding frequency. Additionally, the low-marsh
genotype generally formed longer rhizomes, shoots and leaves, regardless of flooding frequency (Fig. 1 & 2). We argue that these findings yield important implications for our understanding of ecosystem resilience to SLR as well as plant-species distribution in salt marshes.

We found a higher total biomass of the low-marsh genotype, which was particularly pronounced under high flooding frequency (Fig. 1a). This result indicates a better adaptation of the low-marsh genotype to lower elevated, more frequently flooded
conditions. When separating above- and belowground biomass, the high-marsh genotype showed a decrease of both biomass parameters with increasing flooding frequency. Aboveground biomass is important for maintaining photosynthesis (Johnson, 2016), so that its reduction can be interpreted as reduction in performance as well. In contrast to the high-marsh genotype showing a marked reduction in aboveground biomass, the low-marsh genotype maintained aboveground biomass across all flooding treatments.

We also found a decrease of belowground biomass with increasing flooding frequency in both genotypes. This can be interpreted as an adaptive trait, because a reduction of belowground biomass reduces the number of respiring roots and thereby improves the diffusion of oxygen to the roots (Naidoo and Naidoo, 1992; Voesenek et al., 1988). In line with these results, an increased aboveground biomass production while belowground biomass decreased was found for other flooding adapted plant species such as *Taxodium distichum, Danthonia montevidensis* and *Paspalum dilatatum* (Megonigal and Day, 1992; Rubio et

al., 1995).

The difference in aboveground biomass response between the two genotypes seems to be mainly explained by genotype-specific increases in leaf and shoot length, whereas the number of both remained similar (Fig. 2, Table 1). Likewise, Voesenek et al. (1988) found a marked increase in leaf length in the flooding adapted *Rumex palustris* under waterlogging, but no increase in number of leaves. The distinct increase in leaf and shoot length in addition to the simultaneous reduction of belowground

biomass of the low-marsh genotype of *Elymus athericus* found in our study, indicate resource allocation as response to flooding. For other species it was found that reallocated resources fuel elongation of shoots and leaves to maintain gas exchange and avoid light dissipation through water (Blanch et al., 1999; Grace, 1989). Our results suggest that this response may be also present in *E. athericus*, which could improve its chances of survival under higher flooding frequencies e.g. due to accelerated SLR.

Vertical accretion in the minerogenic salt marshes of the Wadden Sea is primarily driven by sedimentation (Allen, 2000; Nolte et al., 2013b), which is strongly controlled by the sediment-trapping capacity of the aboveground biomass (Morris et al., 2002; Yang, 1998). The strong aboveground biomass response to increased flooding frequencies of the low-marsh genotype found in our study may therefore have a positive effect on vertical accretion rates and thereby marsh resilience to rising sea levels.

*E. athericus* is not the only salt-marsh species characterised by a high degree of genetic diversity. In previous studies, genotypes

of several salt-marsh grasses have been described and tested for intraspecific differences in plant response to changing environmental conditions, including *Puccinellia maritima*, *Phragmites australis* and *Spartina alterniflora* (Gray, 1985; Mozdzer and Megonigal, 2012; Proffitt et al., 2003; Seliskar et al., 2002). They showed high genotypic variations affecting colonisation success, species composition and even ecosystem function.

Compared to the root biomass of both genotypes, which responded similarly and decreased with increasing flooding frequency,

rhizome length differed significantly between the genotypes (Table 1). The formation of longer rhizomes by the low-marsh genotype, especially under high flooding frequency, could serve as an escape strategy to expand into more favourable habitats (Hartnett and Bazzaz, 1983; Lovett-Doust, 1981). In previous studies, it was reported that *E. athericus* usually expands via a 'phalanx' growth strategy, which means that parental plants invest in many but rather short rhizomes to utilise resources in a favourable habitat (Bockelmann and Neuhaus, 1999). Field observations of the same authors and results of our study, however,

indicate that *E. athericus* is able to alter its strategy to the 'guerrilla' form, by producing longer rhizomes. The 'guerrilla' strategy is usually found in plants characteristic for early successional stages as it enables plants to spread quickly and exploit new favourable areas (Lovett-Doust, 1981). However, overall rhizome length of our study should be interpreted with caution because of potential edge effects caused by the experimental mesocosms.

The change of expansion strategy together with a better adaptation to higher flooding frequencies may lead to a displacement of the high-marsh genotype under accelerated SLR. However, until now, the Wadden Sea salt marshes are able to cope with current rates of sea level rise due to high accretion rates (Esselink et al., 2017; Nolte et al., 2013a). If rates of SLR remain stable, the low-marsh genotype of the tall grass *E. athericus* has the potential to expand further into the low marsh and outcompete other species via light competition, potentially reducing local species diversity.

## 5 Methodological considerations

We suggest that the experimental setup including a tidal tank and steps proved suitable to investigate the effects of different flooding frequencies on salt-marsh vegetation. Nevertheless, we recommend repeating this experiment in situ, for example as a transplant experiment, to estimate actual effect size under more natural conditions, as drainage and plant-soil interactions might have been different in the tidal tank and could have affected biomass production.

## 6 Conclusion

The present work revealed marked differences in the plant biomass response to changes in flooding frequency between two genotypes of the dominant European salt-marsh grass *Elymus athericus*. Furthermore, we observed large differences in rhizome production between genotypes, which is interpreted as a change in growth strategy. The alteration of its growth strategy and the higher aboveground biomass productivity of the low-marsh genotype implies a larger potential of the low-marsh genotype to invade and establish at lower elevations of the tidal frame. Considering the generally low plant species diversity of salt marshes (e.g. Silliman, 2014; Wanner et al., 2014) and the strong feedbacks between plant growth and accelerated SLR (Kirwan and Megonigal, 2013), it is possible that intraspecific variation and adaptive capacity in salt-marsh plants acts as an important but overlooked mediator of ecosystem resilience.

## 7 Data availability

All data presented in this paper is available from the corresponding author upon request.

## 8 Author contribution

All authors contributed to the design of the experiments. SR and HT conducted the experiment and performed the measurements. SR analysed the data and wrote the manuscript. All authors contributed to the discussion of results and the final manuscript.

## 9 Competing interests

The authors declare that they have no conflict of interest.

## 10 Acknowledgements

We would like to thank Chris Smit and his colleagues from the University of Groningen for the provision of the plants. Furthermore, we would like to acknowledge Max Beiße, Marion Klötzl, and Maren Winnacker for their assistance during the preparation of the experiment. Christoph Reisdorff provided advice on experimental setup and measurements.

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

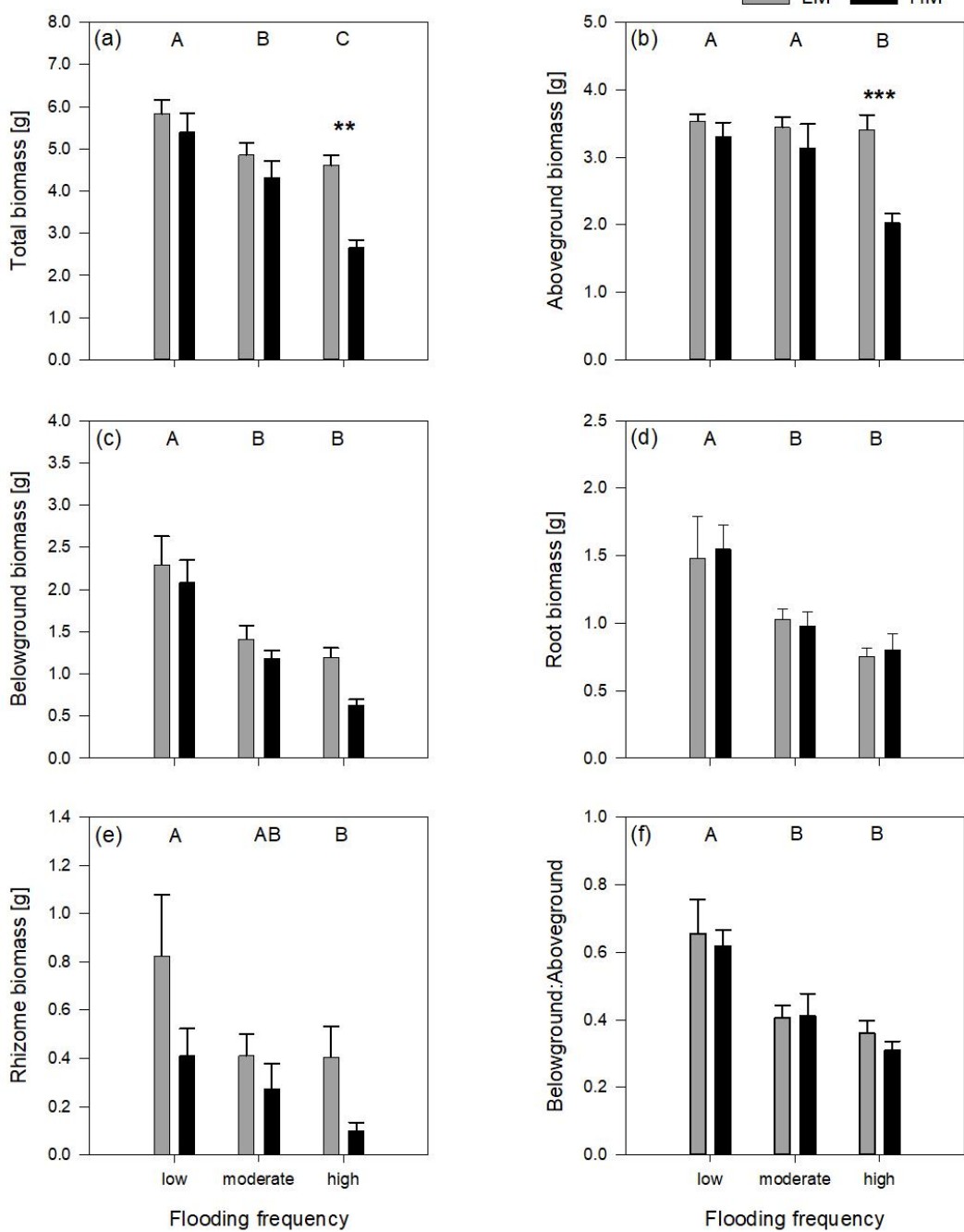

**Figure 1: Total biomass (a), aboveground biomass (b), belowground biomass (c), root biomass (d), rhizome biomass (e) and belowground-aboveground ratio (f) of both genotypes on three different flooding frequencies (mean + standard errors). Stars show significant differences between low-marsh (LM) and high-marsh (HM) genotypes within the same flooding treatment based on**

Tukey's HSD post-hoc test (* p < 0.05; **p < 0.01; *** p < 0.001). Capital letters indicate significant differences between flooding frequencies. For statistics see Table 1.

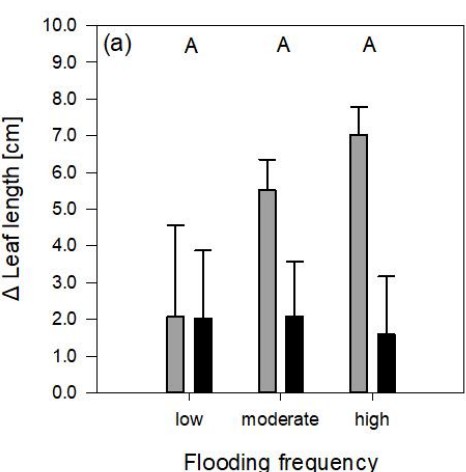 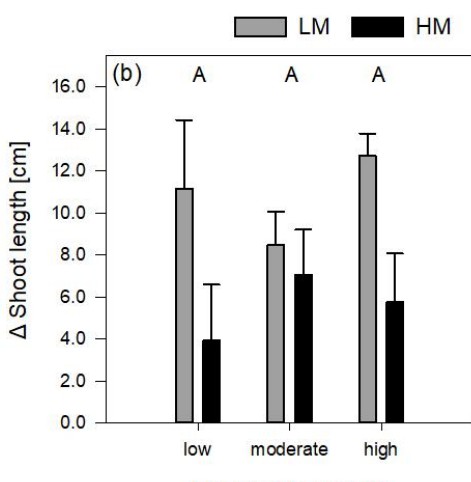

**Figure 2: Delta leaf length (a) and delta shoot length (b) of both genotypes on three different flooding frequencies (mean + standard errors). Capital letters indicate significant differences between flooding frequencies. For statistics see Table 1.**




**Table 1: Summary statistics of main response variables in the experiment testing for effects of flooding and genotype of *Elymus athericus* on its biomass and growth. Count variables (number of shoots, number of leaves) are analysed through GLM, all other variables by a two-way factorial ANOVA. Significant p-values are shown in bold letters. Means and standard errors are shown in Figures 1 and 2.**

| Response variable | statistical test | genotype (df =1) test statistic | genotype (df =1) p-value | flooding frequency (df = 2) test statistic | flooding frequency (df = 2) p-value | genotype * flooding frequency (df = 2) test statistic | genotype * flooding frequency (df = 2) p-value |
|---|---|---|---|---|---|---|---|
| *Total biomass [g]* | factorial ANOVA | F = 13.18 | **≤ 0.001** | F = 18.24 | **≤ 0.001** | F = 3.36 | **≤ 0.05** |
| *Aboveground biomass [g]* | factorial ANOVA | F = 13.84 | **≤ 0.001** | F = 6.38 | **≤ 0.01** | F = 4.77 | **≤ 0.05** |
| *Belowground biomass [g]* | factorial ANOVA | F = 4.31 | **≤ 0.05** | F = 21.93 | **≤ 0.001** | F = 0.53 | 0.59 |
| *Δ Shoot length [cm]* | factorial ANOVA | F = 7.77 | **≤ 0.01** | F = 0.32 | 0.73 | F = 1.03 | 0.37 |
| *Δ Leaf length [cm]* | factorial ANOVA | F = 5.09 | **≤ 0.05** | F = 1.08 | 0.35 | F = 1.42 | 0.25 |
| *Δ Number of shoots* | GLM | Wald = 0.00 | 0.95 | Wald = 5.87 | 0.05 | Wald = 0.78 | 0.68 |
| *Δ Number of leaves* | GLM | Wald = 0.44 | 0.51 | Wald = 19.69 | **≤ 0.001** | Wald = 1.60 | 0.45 |
| *Root biomass [g]* | factorial ANOVA | F = 0.03 | 0.86 | F = 10.69 | **≤ 0.001** | F = 0.07 | 0.93 |
| *Rhizome biomass [g]* | factorial ANOVA | F = 6.49 | **≤ 0.05** | F = 3.84 | **≤ 0.05** | F = 0.51 | 0.60 |
| *Rhizome length [cm]* | factorial ANOVA | F = 6.10 | **≤ 0.05** | F = 1.67 | 0.20 | F = 0.45 | 0.64 |
| *Belowground:Aboveground* | factorial ANOVA | F = 0.34 | 0.56 | F = 14.56 | **≤ 0.001** | F = 0.13 | 0.88 |

**Table 2: Mean and standard error (SE) of rhizome length [cm], delta number of leaves and delta number of shoots for different combinations of factors genotype and flooding frequency. LM = low-marsh genotype, HM = high-marsh genotype.**

| genotype | flooding frequency | N | Rhizome length [cm] | | Δ Number of leaves | | Δ Number of shoots | |
|---|---|---|---|---|---|---|---|---|
| | | | mean | SE | mean | SE | mean | SE |
| LM | low | 8 | 63.38 | 15.13 | 15.88 | 2.43 | 7.00 | 0.82 |
| HM | low | 8 | 37.13 | 9.85 | 18.00 | 3.19 | 8.13 | 1.52 |
| LM | moderate | 8 | 45.28 | 10.35 | 21.75 | 2.45 | 9.50 | 0.78 |
| HM | moderate | 8 | 32.81 | 11.76 | 16.88 | 3.42 | 8.50 | 1.65 |
| LM | high | 8 | 45.63 | 18.10 | 10.43 | 1.53 | 6.25 | 1.18 |
| HM | high | 8 | 9.94 | 2.91 | 8.86 | 0.80 | 5.88 | 1.08 |