# Peer review of "Plant genotype determines biomass response to flooding frequency in tidal wetlands"

_Biogeosciences, 2020_

## Referee Comment (RC1) · Anonymous Referee #1 · 24 Jun 2020

The response of saltmarshes to increased flooding is a highly relevant research topic in times of accelerated sea level rise. This manuscript investigated the response of two genotypes of Elymus athericus to different flooding frequencies. They find that the low marsh genotype is better adapted to higher flooding frequency by allocating resources from below- to aboveground biomass. Generally, I think this is a novel and well-written paper with convincing results and significant implications. This paper likely inspires more research on how genetic effects and evolution of plant species may shape the fate of saltmarshes under SLR, which to my knowledge is currently missing in this field.

Besides, I have some minor comments for improvement, as listed below.

[Figure]

Introduction:

Line 34: '...because their aboveground biomass reduces water flow velocity...' marsh plants facilitate sediment settlement, not only by reducing flow velocity, but also through damping waves. Moreover, references are needed here.

Line 48-49: 'highly species-specific depended', 'depended' should be 'dependent'

Line 53: 'However, such studies on .... community level', references are needed here.

Methods:

Regarding the experimental set-up, more details are needed.

Line 83: It is not clear to me how you transplanted plants from trays to the pots. How many pots were there in total and how many plants per pot? What were the inundation depths for different steps? Please also provide the reason for choosing these three flooding frequency treatments.

Discussion: Line 192: You put it here as '4.1', but there is no '4.2' , '4.3' etc..

What I am missing from the discussion is the implications beyond the species Elymus athericus. How common is genetic variation of saltmarsh plants? Are there other examples that shows marsh plants adapt to changing environment via genetic change/evolution? Moreover, I think the consequences of changing biomass allocation of Elymus athericus for saltmarsh accretion and its response to SLR should also be discussed.

Figure.2 Regarding the results of the post hoc tests (stars), it seems that only Figure. 2a has shown where the difference is significant. For the rest subfigures such as Fig.2 c & d, no stars are added, yet there are obvious differences between the two genotypes for the high flooding frequency treatment.

---

## Referee Comment (RC2) · Anonymous Referee #2 · 25 Jun 2020

Considering the heightened vulnerability of tidal marshes to SLR, an increasing number of studies are examining flooding and other climate change impacts to marsh plant growth and viability and their feedbacks to marsh elevation and resilience. As the authors note, most of these do not consider responses of different genotypes of the same species, but rather responses at the species level or among species. Thus, this experiment, which investigated biomass responses of different plant genotypes to increasing flooding frequency, fills an important gap. While the overall conclusion that the low-marsh genotype is better adapted to flooding than the high-marsh genotype is an intuitive one, this paper provides some direct evidence of biomass responses and suggests that formation of longer rhizomes by the low-marsh genotype serves as a

flooding escape strategy. The paper is generally well-written and presents vegetative response data clearly and succinctly. However, there are several areas in need of attention, as detailed below.

Broader context: Situating this work within the context of other studies examining population-level or genotypic differences in species' responses to flooding/elevation, salinity, nutrient enrichment or other global environmental changes would be helpful and would allow a more robust discussion of the potential implications of genotype-specific differences for ecosystem function and resilience (e.g., Lessmann et al. 1999; Proffitt et al. 2003, 2005; Mozdzer and Megonigal 2012).

Materials and methods: The paper is significantly lacking in important information on the experimental set-up and methodologies, on everything from plant collection, marsh organ construction and maintenance, and the specific measurements (as noted by section below).

Section 2.1: How were the plants collected from the field? Were they intact sods of soil and vegetation? Were they rinsed of site soils before planting? How were they planted and grown in the trays (under what hydro-edaphic conditions, temperature, light availability, density, etc.)? How was plant size determined and standardized across treatments for use in the study (or randomized if standardization not possible)? Although there were some measures of change to account for potential initial differences, additional discussion of how plant size varied (or not) and what efforts were made to control for these differences is warranted; otherwise, subsequent biomass results could be skewed based on differences in initial weights of plants used in the study. What are the soils like at the field site and were they sieved to remove belowground biomass before being used in the pots?

Section 2.2: How were the mesocosms constructed and how did this affect the way in which water filled and drained the pots (were there holes in the bottom so that they filled and drained from below)? How were marsh organs oriented to control for shading

or other effects? Were you limited to 3 flooding levels due to tidal tank size? What was the height difference among steps in the marsh organs and by how much was the marsh surface flooded for each of the treatments? What was the flooding range relative to the mesocosm position; were all pots fully drained at "low tide" or not? How do the flooding treatments compare to the elevations and flooding ranges in the field? Did the flooding treatments encompass the current marsh elevation/flooding gradient or was the study designed to simulate increased flooding as expected with SLR? How did the salinity regime compare to those at the field site? What it the typical growing season for these plants (is 12-weeks a reasonable study length given this marsh's latitude)?

Section 2.3: Were there any hydro-edaphic variables measured? These could confirm treatment effects and help explain observed differences among flooding treatments. Was there any evidence that the plants were nutrient-limited? Did they become "root-bound" over the course of the study?

Results & Discussion: One of the main points made is that flooding leads to shifts in biomass allocation from below- to aboveground for the low-marsh, but not the high-marsh, genotype, but the data presented do not explicitly demonstrate shifts in allocation along the flooding gradient. Why not calculate the root:shoot for both genotypes to test this explicitly?

In the introduction, the authors note different mechanisms of plant-mediated feedbacks to elevation – sediment trapping aboveground and contributions to soil volume belowground. Some discussion of this in light of the results would strengthen the paper. For instance, what are the implications of declining aboveground biomass (for both genotypes) with increased flooding for marsh resilience to SLR? How reliant on sediment accretion are these marshes, and to what extent would reduced sediment trapping capacity be expected to reduce resilience? What about the relative importance of reduced belowground inputs to soil volume in these marshes?

Regarding the conclusion that there is potential for the low-marsh genotype to invade

lower elevations, it would also be worth discussing its adaptability to SLR and its potential to displace the high-marsh genotype as water levels rise. Given that, what are the implications for marsh resilience?

Some additional technical corrections are provided below:

Line 19: "with "increasing flooding frequency."

Lines 37-38: "and often depends on"

Line 52: "if SLR-induced shifts . . . composition also are"

Line 122: introduce LM and HM abbreviations earlier

Line 126: "remained constant"

Line 165: "parameters with increasing flooding frequency."

Lines 172, 175: italicize scientific names

Line 182: "responded similarly and decreased with increasing flooding frequency"

---

## Author Comment (AC1) · 7 Sep 2020

Firstly, we would like to thank **both** reviewers for their time and constructive comments. In the following, we will respond to every comment separately (font in blue) while referring to original line numbers. Sentences that have been changed or added are written in bold letters.

**Anonymous Referee #1**

The response of saltmarshes to increased flooding is a highly relevant research topic in times of accelerated sea level rise. This manuscript investigated the response of two genotypes of Elymus athericus to different flooding frequencies. They find that the low marsh genotype is better adapted to higher flooding frequency by allocating resources from below- to aboveground biomass. Generally, I think this is a novel and well-written paper with convincing results and significant implications. This paper likely inspires more research on how genetic effects and evolution of plant species may shape the fate of saltmarshes under SLR, which to my knowledge is currently missing in this field.
Besides, I have some minor comments for improvement, as listed below.

Thank you very much for your feedback!

Introduction:

Line 34: ': : :because their aboveground biomass reduces water flow velocity: : :' marsh plants facilitate sediment settlement, not only by reducing flow velocity, but also through damping waves. Moreover, references are needed here.

Thank you for pointing this out. We will add that missing information and references as follows to the sentence starting in line 33:

During this process, salt-marsh plants act as 'ecosystem engineers' because their aboveground biomass reduces water flow velocity **and hydrodynamic forces, which results in a decrease in the sediment-loading capacity of the water and an increase in sediment settlement (Morris 2002, Yang 1998).**

Line 48-49: 'highly species-specific depended', 'depended' should be 'dependent' Will be changed.

Line 53: 'However, such studies on : : :. community level', references are needed here. Will be changed.

Methods:

Regarding the experimental set-up, more details are needed.

We agree that the Material and Methods section is rather short. As reviewer two is also missing a more detailed description, we will add more information regarding the extraction and potting of the plants as well as the flooding regime and overall experimental set-up. For more details please also see comments below.

Line 83: It is not clear to me how you transplanted plants from trays to the pots. How many pots were there in total and how many plants per pot?

We transferred single plants to individual pots. We described in line 91, that 'Eight replicates per genotype were placed on each step, so that a total number of 48 plants were used in this study' but we will edit that sentence and distinguish between terms like pots, plants and replicates more

carefully. As reviewer two requested more details regarding the same paragraph (2.1. Elymus athericus) as well, we will change almost the whole paragraph as follows:

**2.1 *Elymus athericus**

Plants were collected in April 2015 from a salt marsh on the Dutch Island Schiermonnikoog (53°30'N, 6°16'E) from stands that have previously been identified to be dominated by genetically distinct populations of *E. athericus*, i.e. high-marsh **(HM)** and low-marsh **(LM)** genotypes (Bockelmann et al., 2003). **Plants and soil were extracted in the form of intact sods to keep them alive during transport. In Hamburg, soil was removed and roots were rinsed before both genotypes were planted separately in trays with standardised potting soil. Until the start of the experiment (i.e. for 24 months), plants were kept under identical environmental conditions in a common garden at the Institute of Plant Science and Microbiology. Ramets of these plants were used for this study. In July 2017, single plants of similar size were transplanted to separate pots and randomly assigned to the flooding treatments. The pots were 15 cm in diameter, 17 cm in height and had holes for drainage in the bottom. They were filled with salt marsh soil taken from the salt marsh at Sönke-Nissen-Koog, Germany (54°36'N, 8°49'E) which was sieved (with a 1 cm mesh) and homogenised beforehand (see Nolte et al. (2013) for soil properties). Eight replicates (i.e. single plants in separate pots) per genotype were placed on each step, so that a total number of 48 plants were used in this study.**

What were the inundation depths for different steps?

Steps in the tidal tank were 20 cm high and flooding reached 3 cm above the respective soil surface. If flooding of the middle or highest step took place, steps of lower elevation were flooded, too. In that case, step height needs to be added to the 3 cm to obtain inundation depth of the lowest or middle step for moderate or low flooding frequency events.

We will add following to the sentence starting in line 86:
Flooding with the respective maximum water level **reached 3 cm above soil surface,** lasted two hours and took place twice a day.

Additionally, we will add the step height to the first sentence of section *2.2 Experimental set-up* (line 83):
Plants were placed onto three steps **(step height: 20 cm)** within a tidal-tank facility (Hanke, Ludewig, & Jensen, 2015) to create three different flooding frequencies.

Please also provide the reason for choosing these three flooding frequency treatments.

We agree that it might be of interest to explain why we chose these flooding frequencies.

We will add following sentences (in bold) to line 89:
The lowest step was flooded every day, which represented the highest frequency. The flooding of the middle step (moderate flooding frequency) happened weekly, while plants on the highest step were flooded only every two weeks (lowest flooding frequency). **Highest and lowest flooding frequency reflect the natural flooding gradient between pioneer zone and high marsh in many NW European salt marshes, including the site where our plants were collected (Bockelmann et al. 2002).** A CTD diver combined with a baro diver (Van Essen Instruments, Delft, The Netherlands) was used to monitor flooding cycles.

Discussion:

Line 192: You put it here as '4.1', but there is no '4.2' , '4.3' etc.. Will be changed.

What I am missing from the discussion is the implications beyond the species Elymus athericus. How common is genetic variation of saltmarsh plants? Are there other examples that shows marsh plants adapt to changing environment via genetic change/evolution? Moreover, I think the consequences of changing biomass allocation of Elymus athericus for saltmarsh accretion and its response to SLR should also be discussed.

Good point. We tried to implement your suggestion and added the following sentences and references to the discussion:

In line 179: …avoid light dissipation through water (Blanch et al., 1999; Grace, 1989). **Our results suggest that this response may be also present in *E. athericus*, which could improve its chances of survival under higher flooding frequencies e.g. due to accelerated SLR.**
**Vertical accretion in the minerogenic salt marshes of the Wadden Sea is primarily driven by sedimentation (Allen 2000, Nolte et al. 2013b), which is strongly controlled by the sediment-trapping capacity of the aboveground biomass (Yang, 1998, Morris et al., 2002). The strong aboveground biomass response to increased flooding frequencies of the low-marsh genotype found in our study may therefore have a positive effect on vertical accretion rates and thereby marsh resilience to rising sea levels.**
***E. athericus* is not the only salt-marsh species characterised by a high degree of genetic diversity. In previous studies, genotypes of several salt-marsh grasses has been described and tested for intraspecific differences in plant response to changing environmental conditions, including *Puccinellia maritima*, *Phragmites australis* and *Spartina alterniflora* (Gray, 1985; Mozdzer and Megonigal, 2012; Seliskar et al., 2002; Proffitt et al., 2003). They showed high genotypic variations affecting colonisation success, species composition and even ecosystem function.**

Additionally, we will add the following sentence at the end of the conclusion in line 204:
**Considering the generally low plant species diversity of salt marshes (e.g. Wanner et al. 2014; Silliman 2014) and the strong feedbacks between plant growth and accelerated SLR (Kirwan and Megonigal 2013), it is possible that intraspecific variation and adaptive capacity in salt marsh plants acts as an important but overlooked mediator of ecosystem resilience.**

Figure.2 Regarding the results of the post hoc tests (stars), it seems that only Figure.
2a has shown where the difference is significant. For the rest subfigures such as Fig.2 c & d, no stars are added, yet there are obvious differences between the two genotypes for the high flooding frequency treatment.

Please note, we used standard error, not standard deviation. We repeated the statistical analyses to double check. There are no other significant differences.

**Anonymous Referee #2**

Considering the heightened vulnerability of tidal marshes to SLR, an increasing number of studies are examining flooding and other climate change impacts to marsh plant growth and viability and their feedbacks to marsh elevation and resilience. As the authors note, most of these do not consider responses of different genotypes of the same species, but rather responses at the species level or among species. Thus, this experiment, which investigated biomass responses of different plant genotypes to increasing flooding frequency, fills an important gap. While the overall conclusion that the low-marsh genotype is better adapted to flooding than the high-marsh genotype is an intuitive one, this paper provides some direct evidence of biomass responses and suggests that formation of longer rhizomes by the low-marsh genotype serves as a flooding escape strategy. The paper is generally well-written and presents vegetative response data clearly and succinctly. However, there are several areas in need of attention, as detailed below.

Broader context: Situating this work within the context of other studies examining population-level or genotypic differences in species' responses to flooding/elevation, salinity, nutrient enrichment or other global environmental changes would be helpful and would allow a more robust discussion of the potential implications of genotypespecific differences for ecosystem function and resilience (e.g., Lessmann et al. 1999; Proffitt et al. 2003, 2005; Mozdzer and Megonigal 2012).

*Thank you very much for your time and your constructive feedback. We will try to improve the discussion by referring to suggested (and other) studies focusing on intraspecific differences of salt-marsh vegetation to changing environments and/or stressors.*

Materials and methods: The paper is significantly lacking in important information on the experimental set-up and methodologies, on everything from plant collection, marsh organ construction and maintenance, and the specific measurements (as noted by section below).

*This observation is in accordance with the comments provided by reviewer one. We will add more information to the whole Material & Methods section.*

Section 2.1: How were the plants collected from the field? Were they intact sods of soil and vegetation? Were they rinsed of site soils before planting? How were they planted and grown in the trays (under what hydro-edaphic conditions, temperature, light availability, density, etc.)? How was plant size determined and standardized across treatments for use in the study (or randomized if standardization not possible)? Although there were some measures of change to account for potential initial differences, additional discussion of how plant size varied (or not) and what efforts were made to control for these differences is warranted; otherwise, subsequent biomass results could be skewed based on differences in initial weights of plants used in the study. What are the soils like at the field site and were they sieved to remove belowground biomass before being used in the pots?

*We agree that more information should be provided here to indicate to the reader that the experiment was conducted most carefully. We will try to answer all of the raised questions and implement them in the paragraph as below (2.1 Elymus athericus paragraph).*
*Furthermore, we tested initial shoot length and number of shoots for differences between genotypes and flooding frequency. There were no significant differences detected.*
*We will add this information later to the discussion to reinforce our assumptions regarding biomass results (in line 167):*
**Initial shoot length and shoot number was tested for differences between genotypes and flooding frequencies to ensure that results were not biased by unequal plant size at the beginning of the experiment. There were no significant differences regarding shoot length (genotype: F = 0.787, p = 0.380; flooding frequency: F = 0.127, p = 0.881; genotype\*flooding frequency: F = 0.231, p = 0.795)**

**and number of shoots (genotype: Wald = 2.203, p = 0.137; flooding frequency: Wald = 0.357, p = 0.837; genotype\*flooding frequency: Wald = 0.005, p = 0.997).**

**2.1 *Elymus athericus**

Plants were collected in April 2015 from a salt marsh on the Dutch Island Schiermonnikoog (53°30'N, 6°16'E) from stands that have previously been identified to be dominated by genetically distinct populations of *E. athericus*, i.e. high-marsh **(HM)** and low-marsh **(LM)** genotypes (Bockelmann et al., 2003). **Plants and soil were extracted in the form of intact sods to keep them alive during transport. In Hamburg, soil was removed and roots were rinsed before both genotypes were planted separately in trays with standardised potting soil. Until the start of the experiment (i.e. for 24 months), plants were kept under identical environmental conditions in a common garden at the Institute of Plant Science and Microbiology. Ramets of these plants were used for this study. In July 2017, single plants of similar size were transplanted to separate pots and randomly assigned to the flooding treatments. The pots were 15 cm in diameter, 17 cm in height and had holes for drainage in the bottom. They were filled with salt marsh soil taken from the salt marsh at Sönke-Nissen-Koog, Germany (54°36'N, 8°49'E) which was sieved (with a 1 cm mesh) and homogenised beforehand (see Nolte et al. (2013) for soil properties). Eight replicates (i.e. single plants in separate pots) per genotype were placed on each step, so that a total number of 48 plants were used in this study.**

Section 2.2: How were the mesocosms constructed and how did this affect the way in which water filled and drained the pots (were there holes in the bottom so that they filled and drained from below)?

Details regarding the pots will be added to section 2.1. (see above):
**The pots were 15 cm in diameter, 17 cm in height and had holes in the bottom.**

How were marsh organs oriented to control for shading or other effects?

The tidal tank was north orientated, shading was very little at the back end of the middle and lowest step but we circulated pots at least once a week to minimise possible effects (further described in line 92).

Were you limited to 3 flooding levels due to tidal tank size?

Yes, the size of the tank is limited.

What was the height difference among steps in the marsh organs and by how much was the marsh surface flooded for each of the treatments?

The steps were 20 cm high. Water level of respective maximum flooding reached 3 cm above soil surface. That means in the case of the lowest flooding frequency (= flooding of the highest step), plants on the middle step experienced an inundation depth of 23 cm above soil surface while plants on the lowest step were completely under water as water level was 43 cm above soil surface. According to this, on moderate flooding frequency, water level reached 3 cm and 23 cm above soil surface of plants standing on the middle and lowest step respectively.

We will add the following to the sentence starting in line 86:
Flooding with the respective maximum water level **reached 3 cm above soil surface,** lasted two hours and took place twice a day.

Additionally, we will add the step height to the first sentence of section *2.2 Experimental set-up* (line 83):
Plants were placed onto three steps **(step height: 20 cm)** within a tidal-tank facility (Hanke, Ludewig, & Jensen, 2015) to create three different flooding frequencies.

What was the flooding range relative to the mesocosm position; were all pots fully drained at "low tide" or not?

Minimum water level was approx. 50 cm below the bottom edge of the pots standing on the lowest step, so all pots were fully drained between flooding events. Difference between minimum and maximum water level was approx. 110 cm.

We will add following to line 86: …between three different maximum water levels. **Pots were fully drained between flooding events.** Flooding with the respective maximum…

How do the flooding treatments compare to the elevations and flooding ranges in the field?
Did the flooding treatments encompass the current marsh elevation/flooding gradient or was the study designed to simulate increased flooding as expected with SLR?

The flooding gradient in our experiment covered natural flooding conditions from the pioneer zone to the high marsh of Schiermonnikoog (where the genotypes used in this study originate from).

We will add following sentences (in bold) to line 89:
The lowest step was flooded every day, which represented the highest frequency. The flooding of the middle step (moderate flooding frequency) happened weekly, while plants on the highest step were flooded only every two weeks (lowest flooding frequency). **Highest and lowest flooding frequency reflect the natural flooding gradient between pioneer zone and high marsh in many NW European salt marshes, including the site where our plants were collected (Bockelmann et al. 2002).** A CTD diver combined with a baro diver (Van Essen Instruments, Delft, The Netherlands) was used to monitor flooding cycles.

How did the salinity regime compare to those at the field site?

Salinity of coastal waters close to salt marshes in NW Europe can vary between approx. 15 – 30 ppt, so we chose the average.

What it the typical growing season for these plants (is 12-weeks a reasonable study length given this marsh's latitude)?

Growing season of *Elymus athericus* is approx. from end of March until end of October. Unfortunately, we had a rather cold spring in 2017 so we decided to give the plants more time to develop.

Section 2.3: Were there any hydro-edaphic variables measured? These could confirm treatment effects and help explain observed differences among flooding treatments. Was there any evidence that the plants were nutrient-limited? Did they become "rootbound" over the course of the study?

We did not measure hydro-edaphic variables but at the end of the experiment, plants were neither rootbound nor showed any sign of chlorosis due to nutrient limitation.

Results & Discussion: One of the main points made is that flooding leads to shifts in biomass allocation from below- to aboveground for the low-marsh, but not the high marsh, genotype, but the data presented do not explicitly demonstrate shifts in allocation along the flooding gradient. Why not calculate the root:shoot for both genotypes to test this explicitly?

We indeed tested for effects on the root:shoot ratio. Root:shoot ratio differed significantly between genotypes and flooding frequency. The interaction of both factors was not significant (genotype: F = 4.453, p < 0.05; flooding frequency: F = 5.869, p < 0.01; genotype*flooding frequency: F = 1.240, p > 0.05).
We will add details to sections Material & Methods and Results and add F- and p-values to table 1 (see below).
Despite the statistically insignificant interaction term of flooding and genotype, one can see a tendency toward different flooding-response patterns of the two genotpyes in the figure below. For the initial submission we wanted to focus on the fact that differences between genotypes are driven by the strong aboveground response and, therefore, did not show this figure. If reviewers and editors would like to see it in the manuscript we would, of course, be happy to include it.

Information that will be added:
In line 99 (Material and Methods): **Root biomass (belowground biomass without rhizomes) and aboveground biomass was used to calculate root-shoot ratio.**
In line 132 (Results): **Root-shoot ratio was significantly affected by genotype and flooding frequency but the interaction was not significant (Table 1). Mean root-shoot ratio of low- and high-marsh genotypes differed the most under highest flooding frequency (LM: 0.22 ± 0.06, HM: 0.39 ± 0.12), although the post-hoc test did not detect a significant difference.**

[Figure]

Figure 1: Root-Shoot ratio of both genotypes under three different flooding frequencies (mean+ standard error).

In the introduction, the authors note different mechanisms of plant-mediated feedbacks to elevation – sediment trapping aboveground and contributions to soil volume belowground.
Some discussion of this in light of the results would strengthen the paper. For instance, what are the implications of declining aboveground biomass (for both genotypes) with increased flooding for marsh resilience to SLR? How reliant on sediment accretion are these marshes, and to what extent

would reduced sediment trapping capacity be expected to reduce resilience? What about the relative importance of reduced belowground inputs to soil volume in these marshes?
Regarding the conclusion that there is potential for the low-marsh genotype to invade lower elevations, it would also be worth discussing its adaptability to SLR and its potential to displace the high-marsh genotype as water levels rise. Given that, what are the implications for marsh resilience?

Thank you for these great suggestions. We will be adding the following to the discussion:

In line 179: …avoid light dissipation through water (Blanch et al., 1999; Grace, 1989). **Our results suggest that this response may be also present in *E. athericus*, which could improve its chances of survival under higher flooding frequencies e.g. due to accelerated SLR.**
**Vertical accretion in the minerogenic salt marshes of the Wadden Sea is primarily driven by sedimentation (Allen 2000, Nolte et al. 2013b), which is strongly controlled by the sediment-trapping capacity of the aboveground biomass (Yang, 1998, Morris et al., 2002). The strong aboveground biomass response to increased flooding frequencies of the low-marsh genotype found in our study may therefore have a positive effect on vertical accretion rates and thereby marsh resilience to rising sea levels.**
***E. athericus* is not the only salt-marsh species characterised by a high degree of genetic diversity. In previous studies, genotypes of several salt-marsh grasses has been described and tested for intraspecific differences in plant response to changing environmental conditions, including *Puccinellia maritima*, *Phragmites australis* and *Spartina alterniflora* (Gray, 1985; Mozdzer and Megonigal, 2012; Seliskar et al., 2002; Proffitt et al., 2003). They showed high genotypic variations affecting colonisation success, species composition and even ecosystem function.**

Additionally we will add the following to line 191:
**The change of expansion strategy together with a better adaptation to higher flooding frequencies may lead to a displacement of the high-marsh genotype under accelerated SLR. However, until now, the Wadden Sea salt marshes are able to cope with current rates of sea level rise due to high accretion rates (Nolte et al., 2013b; Esselink et al. 2017). If rates of SLR remain stable, the low-marsh genotype of the tall grass *E. athericus* has the potential to expand further into the low marsh and outcompete other species via light competition, potentially reducing local species diversity.**

Furthermore, the following sentence will be added at the end of the conclusion in line 204:
**Considering the generally low plant species diversity of salt marshes (e.g. Wanner et al. 2014; Silliman 2014) and the strong feedbacks between plant growth and accelerated SLR (Kirwan and Megonigal 2013), it is possible that intraspecific variation and adaptive capacity in salt marsh plants acts as an important but overlooked mediator of ecosystem resilience.**

Some additional technical corrections are provided below:
Line 19: "with "increasing flooding frequency." Will be changed.
Lines 37-38: "and often depends on" Will be changed.
Line 52: "if SLR-induced shifts : : : composition also are" Will be changed.
Line 122: introduce LM and HM abbreviations earlier Will be changed.
Line 126: "remained constant" Will be changed.
Line 165: "parameters with increasing flooding frequency." Will be changed.
Lines 172, 175: italicize scientific names Will be changed.
Line 182: "responded similarly and decreased with increasing flooding frequency" Will be changed.

**References**

Allen, J. R. L.: Morphodynamics of Holocene salt marshes : a review sketch from the Atlantic and Southern North Sea coasts of Europe, Quat. Sci. Rev., 19, 1155–1231, 2000.

Bockelmann, A. C., Bakker, J. P., Neuhaus, R. and Lage, J.: The relation between vegetation zonation, elevation and inundation frequency in a Wadden Sea salt marsh, Aquat. Bot., 73(3), 211–221, doi:10.1016/S0304-3770(02)00022-0, 2002.

Esselink, P., Duin, W. E. Van, Bunje, J., Cremer, J., Folmer, E. O., Frikke, J., Glahn, M., Groot, A. V. De, Hecker, N., Hellwig, U., Jensen, K., Körber, P., Petersen, J. and Stock, M.: Salt marshes, Wadden Sea Qual. Status Rep., Eds.: Kloepper S. et al., Common Wadden Sea Secret, 2017.

Gray, A. J.: Adaptation in perennial coastal plants - with particular reference to heritable variation in Puccinellia maritima and Ammophila arenaria*, Vegetatio, 61, 179–188, 1985.

Kirwan, M. L. and Megonigal, J. P.: Tidal wetland stability in the face of human impacts and sea-level rise, Nature, 504, 53–60, doi:10.1038/nature12856, 2013.

Morris, J. T., Sundareshwar, P. V., Nietch, C. T., Kjerfve, B. and Cahoon, D. R.: Responses of coastal wetlands to rising sea level, Ecology, 83(10), 2869–2877, 2002.

Mozdzer, T. J. and Megonigal, J. P.: Jack-and-Master Trait Responses to Elevated $CO_2$ and N : A Comparison of Native and Introduced Phragmites australis, PLoS One, 7(10), doi:10.1371/journal.pone.0042794, 2012.

Nolte, S., Müller, F., Schuerch, M., Wanner, A., Esselink, P., Bakker, J. P. and Jensen, K.: Does livestock grazing affect sediment deposition and accretion rates in salt marshes ?, Estuar. Coast. Shelf Sci., 135, 296–305, doi:10.1016/j.ecss.2013.10.026, 2013.

Proffitt, C. E., Travis, S. E. and Edwards, K. R.: Genotype and elevation influence Spartina alterniflora colonization and growth in a created salt marsh, Ecol. Appl., 13(1), 180–192, 2003.

Seliskar, D. M., Gallagher, J. L., Burdick, D. M. and Mutz, L. A.: The regulation of ecosystem functions by ecotypic variation in the dominant plant : a Spartina alterniflora salt-marsh case study, J. Ecol., 90, 1–11, 2002.

Silliman, B. R.: Quick guide: Salt marshes, Curr. Biol., 24(9), 348–350, doi:10.1016/j.cub.2014.03.001, 2014.

Wanner, A., Suchrow, S., Kiehl, K., Meyer, W., Pohlmann, N., Stock, M. and Jensen, K.: Scale matters: Impact of management regime on plant species richness and vegetation type diversity in Wadden Sea salt marshes, Agric. Ecosyst. Environ., 182, 69–79, doi:10.1016/j.agee.2013.08.014, 2014.

Yang, S. L.: The Role of Scirpus Marsh in Attenuation of Hydrodynamics and Retention of Fine Sediment in the Yangtze Estuary, Estuar. Coast. Shelf Sci., 47, 227–233, 1998.

---

## Author Response (AR2)

**Reviewer:**

This manuscript examines responses of different genotypes of the same species to changes in inundation, thereby addressing an understudied area of potential plant responses to SLR. The results focus on plant responses to different flooding frequencies, highlighting differences in biomass, leaf and shoot, and rhizome growth between low-marsh and high-marsh genotypes of E. athericus. The revised version of the paper addresses most of the concerns raised during the first round of review by situating this work within the context of other studies examining population-level or genotypic differences in species' responses to global change factors, providing more information in the methods, and expanding the data presented in the results. The addition of these new data, however, raised some additional questions that should be addressed in a subsequent revision. I've noted these questions and some others below.

We appreciate the reviewer's comments. Below we address each of them separately. Line numbers refer to the newly revised manuscript. The author's reply is in blue font, and additions or changes to the original text are written in bold letters.

Hypotheses: consider expanding to include more than just the biomass response, as several variables deal with other growth responses (leaf and shoot length, rhizome length, etc.) and are subsequently discussed as important findings of the study.

We specified the hypothesis as follows:

> Line 72: To test the hypothesis that the low-marsh genotype performs better at increased flooding frequencies than the high-marsh genotype, **which would be reflected in above- and belowground growth parameters like biomass production and shoot, leaf and rhizome lengths,** we conducted a factorial marsh organ experiment. We exposed both genotypes of *E. athericus* to three different flooding frequencies and quantified plant growth parameters to compare their performance and assess morphological adaptations of the low-marsh genotype.

Materials and methods: The revisions addressed most of the concerns raised previously, but more information on the field site from which plants were collected and how the experimental design relates to field-relevant conditions would be useful, as noted below for the different sections.

Section 2.1. Provide a brief description of the tides and elevation range of the site, the distribution of the genotypes along this gradient, and other environmental data (e.g., salinity), which would then allow the reader to understand how the experimental design corresponds to the field conditions.

In line 81, we would like to refer to the paper of Bockelmann et al. (2003) as it provides a very good description of the low- and high-elevation sites on Schiermonnikoog where the two genotypes were collected.

We added the available information accordingly:

> Line 81: **On Schiermonnikoog, *Elymus athericus* can be found in higher and lower elevated sites, which are inundated 20 – 125 and 90 – 270 times per year, respectively (Bockelmann et al., 2003). Soil salinities range from 22 to 29 ppt (Bolhuis et al., 2013).**

Section 2.1, line 84: How did you quantify "similar size" of individual plants? Was it just initial shoot length and number as noted in the discussion? If so, this does not necessarily reflect potential differences in initial biomass. Are these variables correlated with biomass, and if so, are their allometric relationships to demonstrate this relationship? What about belowground biomass? It

would be helpful to more explicitly note the measures taken to document initial plant variables, which when combined with the random assignment of plants to treatments, minimizes potential size-based bias. If these data are not available, then anecdotal visual assessments of biomass (no obvious outliers), when combined with the shoot and leaf length data and random assignment of plants to treatments, would help reduce concerns about initial size bias in the study.

Additional comment:
Lines 180-185. Move this to the results or methods as evidence that any initial differences in plant size were insignificant.

It was not possible to measure other parameters like biomass at the beginning of the experiment as those measurements would have been destructive, but we will follow the reviewer's suggestion by adding the information regarding the visual assessment as follows:

Line 87: In July 2017, single plants of similar **biomass were selected based on visual assessment (no obvious outliers)**, transplanted to separate pots and randomly assigned to the flooding treatments. **Initial shoot length and shoot number was tested for differences between genotypes and flooding frequencies to ensure that results were not biased by unequal plant size at the beginning of the experiment. There were no significant differences regarding shoot length (genotype: F = 0.787, p = 0.380; flooding frequency: F = 0.127, p = 0.881; genotype\*flooding frequency: F = 0.231, p = 0.795) and number of shoots (genotype: Wald = 2.203, p = 0.137; flooding frequency: Wald = 0.357, p = 0.837; genotype\*flooding frequency: Wald = 0.005, p = 0.997).**

Section 2.3.1. The inclusion of root:shoot is a nice addition, but it raises some new questions. For root:shoot, rhizomes were excluded, yet it appears that rhizomes were included in the belowground biomass values reported in Fig. 1. Based on an estimated calculation using the biomass values in the figure, the root:shoot values do not appear to be derived from the values reported for above- and below-ground biomass in Fig. 1. If rhizomes had been included as belowground biomass for the calculations of root:shoot, it appears that a different pattern for root:shoot would emerge, especially for the high flooding treatment. This raises the question of where and when rhizomes were incorporated into measures of biomass and how this affects interpretations of results. To clarify this confusion, explain what material is included in measures of above- and below-ground biomass in the methods, justify the removal of rhizomes from calculations of root:shoot (or include rhizomes in the calculations and refer to it as belowground:aboveground ratio instead), and consider the implications of not including rhizomes in the calculation of root:shoot. For example, rhizomes would contribute to vertical resilience by contributing to soil volume, a mechanism noted in the paper, and may be an important component when considering potential allocation responses by the genotypes.

We agree with the reviewer that rhizomes also contribute to vertical accretion and should therefore be included in a ratio describing possible allocation patterns. Consequently, we will use the belowground:aboveground ratio instead of root:shoot. Please note, root and rhizome biomass will still be displayed separately (Figure 1d & 1e). Sentences in section 2.3.1 were changed as follows:

Line 116: **Belowground biomass was divided into rhizomes and roots and weighed. The length of rhizomes was also recorded. Additionally, above- and belowground biomass were used to calculate the belowground:aboveground ratio.**

Section 2.3.2. Provide additional information here about the differences in initial measures of plant growth, as noted above.

We added this information to section 2.1. (see comment above).

Results: Consider reorganizing the subsections so that all biomass measures are presented consecutively (Total biomass, above- and belowground biomass, root and rhizome biomass, ratio), followed by shoot and leaf length and number. The figures could be grouped similarly, rather than having the shoot and leaf data combined with the rhizome and root biomass.

In section 3.1, we merged all biomass measures into one paragraph following the suggested order. We also changed figures 1 and 2 accordingly.

Section 3.1. Include reference to Fig. 1 when presenting root:shoot results.

Addressed accordingly.

Section 3.2. Present these data in the order in which they are presented in figure 2 (or reorder the panels in figure 2).

Addressed accordingly.

Sections 3.2-3.3. The data for number of shoots and leaves and rhizome length are not shown anywhere, yet there are significant patterns of interest for leaves and rhizome length that are subsequently highlighted in the discussion. Add a table or figure to show the means for the different genotype x flooding combinations for these data so that all patterns are shown for variables with significant findings.

We added the new Table 2 showing mean and standard error of rhizome length, delta number of leaves and delta number of shoots for different genotype x flooding combinations.

Discussion:

Lines 179-180. What is the significance of low-marsh genotypes maintaining aboveground biomass regardless of flooding treatment?

We wanted to emphasise that in comparison to the high-marsh genotype, the low-marsh genotype did not show a stress response. We rephrased the original sentence to:

> Line 187: **In contrast to the high-marsh genotype showing a marked reduction in aboveground biomass, the low-marsh genotype maintained aboveground biomass across all flooding treatments.**

Starting in line 208, we further describe the importance of aboveground biomass for vertical accretion and marsh resilience.

Lines 180-185. Move this to the results or methods as evidence that any initial differences in plant size were insignificant.

We moved it to section 2.1. (see above).

Update the discussion considering any changes in the presentation and interpretation of root:shoot data.
No changes required because rhizome and root biomass have been discussed irrespective of the belowground:aboveground- or root:shoot-ratio.

Some additional technical corrections are provided below:

Line 16: delete "a" prior to "higher aboveground biomass" - done

Line 41: add a space between "loss" and "(Chmura…" - done

Line 73: close the parentheses – In the original pdf file (bg-2020-manuscript-version3.pdf) we see that the parenthesis is already closed. Or do you want us to remove the parentheses?

Line 85: add "(described in 2.2)" after "flooding treatments" - done

Line 85: add "to facilitate drainage" at the end of the sentence. - done

Line 88: change "placed on each step" to "assigned to one of three flooding treatments" because steps are not explained until the next section. - done

Lines 143-145: cite Fig. 1 - done

Line 151: "with increasing flooding" - done

Line 152: Section numbering is repeated. This should be Section 3.3. - done

Line 156: Should be Section 3.4. - done

Line 178: include citations - done

Line 179: delete "a" before "high aboveground biomass" - done

Line 206: "have" instead of "has" - done

References: check formatting; some title are capitalized throughout and others aren't; scientific names are not italicized - done

Figures: consider adding panel labels (A, B, C, D) and adding the specific panel reference to the text when figures are referenced. Also, check captions that suggest asterisks indicate significance, but no asterisks are included on the graphs.

We removed the sentence regarding the asterisks in the captions of figure 2.

We added panel labels to both figures.

For variables with main effects of flooding but no interactions, these differences can be shown on the graphs by adding letters or symbols denoting differences over the LM-HM pairs for each flooding treatment. Similarly, it would be useful to note variables with differences between genotypes (in the absence of interactive effects) somehow – perhaps by noting this in caption.

Differences between genotypes within a given flooding treatment have already been indicated using asterisks, letters will be added to show showing differences between flooding treatments to figures 1 and 2.

**Editor:**

Comments to the Author:
Dear authors,

Thank you for the revised manuscript and response letter, which have now been seen by one of the previous reviewers. The reviewer is positive about the changes that you made, but made some more comments and suggestions that need to be considered before this manuscript can be accepted for publication.

Especially the root data raised additional questions. Please clarify why you opted to exclude rhizomes from the belowground biomass for calculation of the root:shoot ratio and what the implications are. You can also consider showing the root:shoot ratio both with and without including rhizomes.

In addition, I would like you to rethink the use of ANCOVA for the statistics. This is not only a more powerful technique, but I think it is also a more correct one for your dataset.

Kind regards,

Sara Vicca

We appreciate the thoughtful comments provided by the Editor.

We agree with the reviewer that rhizomes could play an important role for vertical accretion in the studied system and should not be excluded from the calculated ratio. In addition to that, root and rhizome biomass remain separately displayed/discussed. We made changes to the text, figures and tables accordingly.

We compared ANCOVA and ANOVA results (Table 1 below). ANCOVA does not seem to provide more statistical power in our case. Specifically, we found for four parameters significant effects with ANOVA that were not significant with ANCOVA. Importantly, the opposite (i.e. non-significant with ANOVA but significant with ANCOVA) was never true. We believe that this is the case because ANCOVA assumes that the regression relationship between the dependent variable and covariate is linear. This linearity, however, is not given for all our parameters. In fact, the literature on plant-biomass responses to flooding in tidal wetlands usually shows non-linear, unimodal relationships between biomass parameters and flooding (Kirwan and Guntenspergen 2012, Langley et al. 2013, Redelstein et al. 2018, Mueller et al. 2016). We therefore did not hypothesize linearity and used ANOVA instead. As our study design only includes three flooding treatments, any model assuming a non-linear relationship between flooding and plant parameters is not useful either.

**Table 1:** Results (p-values and significance codes) generated by ANOVA and ANCOVA. Significance codes: *** = p < 0.001; ** = p < 0.01; * = p < 0.05. Orange highlights effects that were detected significant by ANOVA but not ANCOVA.

[revised manuscript text omitted]